# Stigma and Glume Characteristics Synergistically Determine the Stigma Exsertion Rate in Thermo-Photo-Sensitive Genic Male Sterile Wheat

**DOI:** 10.3390/plants13162267

**Published:** 2024-08-15

**Authors:** Hongsheng Li, Zhonghui Yang, Shaoxiang Li, Ahmed M. S. Elfanah, Sedhom Abdelkhalik, Xiong Tang, Jian Yin, Mingliang Ding, Kun Liu, Mujun Yang, Xiue Wang

**Affiliations:** 1State Key Laboratory of Crop Genetics & Germplasm Enhancement and Utilization, College of Agriculture, Collaborative Innovation Center for Modern Crop Production (CIC-MCP), Nanjing Agricultural University, Nanjing 210095, China; lhs@yaas.org.cn; 2Food Crops Research Institute, Yunnan Academy of Agricultural Sciences, Kunming 650205, China; yzh@yaas.org.cn (Z.Y.); lsx@yaas.org.cn (S.L.); ahmed.elfanah@arc.sci.eg (A.M.S.E.); sedhom_aiad@yahoo.com (S.A.); txynnkylzs@163.com (X.T.); yinjian199605@163.com (J.Y.); dml@yaas.org.cn (M.D.); lk@yaas.org.cn (K.L.); 3Wheat Research Department, Field Crops Research Institute, Agricultural Research Center, Giza 12619, Egypt

**Keywords:** hybrid wheat, floral architecture, stigma exsertion rate, out-crossing seed setting rate, thermo-photo-sensitive genic male sterile

## Abstract

Wheat hybrids have been widely demonstrated to have remarkable heterosis or hybrid vigor in increasing yield potential and stability since the 1960s. Two-line hybrid wheat can achieve higher yields than local varieties, especially in marginal environments. However, the commercial application of hybrid wheat is hindered by higher seed costs, primarily due to lower yields in hybrid seed production. Stigma exsertion has been verified as a decisive factor in increasing rice’s hybrid seed yield, but more investigation is needed in hybrid wheat breeding and production. In this study, four thermo-photo-sensitive genic male sterile lines, including K41S, K64S, K66S, and K68S, with different stigma exsertion rates (SERs) were used to compare the differences in floral architecture relative to stigma exsertion over two growing seasons. The results revealed that the K41S and K64S exhibited a relatively higher SER at 21.87% and 22.81%, respectively. No exserted stigma was observed in K66S, and K68S had an SER of only 0.82%. This study found that the stigma length, glume width and the length–width ratio of the glume were significantly correlated with the SER, with correlation coefficients of 0.46, −0.46 and 0.60, respectively. Other stigma features such as the branch angle, stretch width and hairbrush length, as well as the glume length, also had a weakly positive correlation with SER (r = 0.09–0.27). For K41S and K64S, the SER was significantly affected by the differences in the stigma branch angle and stigma stretch width among florets. A cross-pollination survey showed that the out-crossing ability of florets with an exserted stigma was about three times as high as that of florets with a non-exserted stigma. As a result, the stigma-exserted florets that accounted for 21.87% and 22.81% of the total florets in K41S and K64S produced 46.80% and 48.53% of the total cross-pollinated seeds in both sterile lines. These findings suggest that a longer stigma combined with a slender glume appears to be the essential floral feature of stigma exsertion in sterile wheat lines. It is expected that breeding and utilizing sterile lines with a higher SER would be a promising solution to cost-effective hybrid wheat seed production.

## 1. Introduction

Wheat (*Triticum aestivum* L.) is an indispensable crop that underpins food security worldwide. It provides more than 20% of the total food calories for the global population. Climate change (e.g., global warming) and the incidence of new diseases and pest pathogens make meeting the demand rate for wheat annually (1.9%) very challenging because the current rate is approximately 0.9% [1,2]. The hybrid wheat breeding approach addresses this problem and enhances the potential and stability of yield, especially in unfavorable/marginal environments [3,4,5,6]. Hybrid wheat yield is a potential advantage compared to the highest inbred commercial cultivars, with the increase ranging from 10 to 20%, especially in saline and rainfed fields [7,8,9,10,11,12].

In autogamous crops such as wheat, inducing male sterility to prevent self-pollination is a crucial method in hybrid production [13,14]. Hybrid wheat breeding programs have exploited several systems since the 1960s to induce male sterility, including the three-line system based on cytoplasmic male sterility (CMS) [15,16,17,18], two-line hybrid wheat systems, e.g., chemical hybridization agents (CHAs) [19,20,21], photoperiod-sensitive cytoplasmic male sterility (PCMS) [22,23], and XYZ-4E-ms [24,25]. In China, another two-line hybrid wheat system using a thermo-photo-sensitive genic male sterile (TPSGMS) line was created in 1992 [6]; this has released 22 hybrid cultivars. Meanwhile, CHA, XYZ-4E-ms and CMS systems have also released ten hybrid cultivars in China (http://www.c-seed.cn, accessed on 1 January 2024). However, hybrid wheat has not achieved large-scale application like hybrid rice or maize due to the higher seed costs caused by a lower yield in hybrid seed production, a common bottleneck for all hybrid wheat systems [9,13,26].

Redesigning male and female idiotypes as parental lines, especially in floral architecture, would help to enhance the out-crossing ability, consequently ensuring an appropriate seed-setting rate in hybrid production [13,14,27]. Thus, the ideal pollinator would be a combiner with more anther extrusion and a good pollen-shedding ability, and such materials are rich in Chinese wheat germplasm [13,28]. On the other hand, the female idiotype entails possessing a specific modification, such as short plants, chasmogamous florets (palea and lemma opening) with long periods of pollen receptivity, a long stigma, long stigmatic hair, and a high stigma exsertion rate. Stigma-exserted florets in sterile lines have been reported to have a better out-crossing ability than glume-opening florets in wheat and rice [26,28,29]. A sterile line with a high stigma exsertion rate (SER) can trap more pollen, improve cross-pollination, and increase the efficiency of hybrid seed production in rice [30,31,32,33,34]. Nevertheless, the contribution of stigma exsertion to the seed-set of hybrid seed production in wheat has been hardly considered so far because few sterile wheat lines possess obvious stigma exsertion. Therefore, it is expected that the development and utilization of sterile lines with a high SER will be a promising way to further uplift the yield of hybrid seed production in wheat.

The TPSGMS line in wheat induces sterility under low-temperature and short-day conditions from the meiosis to monokaryotic stages of pollen development. Meanwhile, the sterile line can multiply by self-pollination under high-temperature and long-day conditions; thus, the maintainer line is circumvented, which simplifies the technology of hybrid seed production. Furthermore, this hybrid breeding strategy provided a practical and appropriate method for heterosis utilization in wheat, with broader restorer germplasm and simple hybrid seed production procedures [35,36,37]. Therefore, the two-line system based on TPSGMS has become the main approach to hybrid wheat breeding in China since 1992 [6]. We initially observed stigma exsertion from a newly developed TPSGMS line K78S in 2000 [38]. After that, dozens of TPSGMS lines derived from K78S were developed, but only a few lines showed stigma exsertion, such as K41S, K63S, K64S, K239S, and the stigma exsertion rate (SER) varied significantly among lines [28,36]. Unfortunately, the difference in the floral architecture between sterile wheat lines with and without stigma exsertion remains unknown. In this study, we identified the characteristics of the floral architecture related to a higher SER in wheat TPSGMS lines by comparing the morphological differences in the floral architecture among sterile lines with and without an exserted stigma and investigated the contribution of stigma-exserted florets to the hybrid seed set.

## 2. Results

### 2.1. Assessment of Sterility and Stigma Exsertion of TPSGMS Lines

The results from pollen staining with 1% I_2_-KI solution are illustrated in Figure 1. During the two growing seasons, on all three sowing dates, most pollens of all sterile lines appeared malformed. All observed pollens could not be stained by the I_2_-KI solution, indicating that they were entirely abortive (sterile). Additionally, all bagged spikes of the four sterile lines from three sowing dates yielded no seed. Consequently, four sterile lines had 100% sterility on three sowing dates during the two growing seasons, which means that all seeds obtained in these sterile lines were from cross-pollination. Meanwhile, pollens of the restorer Yunmai112 were well developed and could be darkly stained, and were therefore fertile.

As a description of the floral architecture attributes, the stigma exsertion rate (SER) was estimated during the three sowing times and both seasons. K41S and K64S clearly showed exposed stigmas, while K68S and K66S had weak or no stigma exsertion (Figure 2).

The mean floret number (TFN) per spike of the four sterile lines ranged from 87.0 to 98.2 florets in both seasons. Therefore, regarding the exserted stigma number (FSEN) and SER, K41S and K64S exhibited a relatively higher SER at 21.87% and 22.81%, respectively, while K66S and K68S had an SER of 0% and 0.82%, respectively, significantly lower than that in K41S and K64S (Table 1).

### 2.2. Floral Architecture, Especially Stigma Structure

The typical characteristics of the stigmas in the four sterile lines are illustrated in Figure 3, which reveals the morphological structure and differences. Each sterile line displayed a different stigma architecture during the flowering stage. The pistil’s shape/volume and stigma growth habits include the stigma length and the angle between the two stigma branches. For example, line K66S has a narrow angle compared to the other three lines. Conversely, K64S has a big pistil volume, with different stigma lengths and hairbrush on the stigma branches.

K66S had the shortest stigma length (SL) at 3.14 mm, which is significantly different from that of the other three lines. K64S showed the longest SL at 4.25 mm. The average SL (4.08 mm) of the lines with a higher SER, namely K41S and K64S, is significantly higher than that (3.67 mm) of the lines with a lower SER, namely K66S and K68S. The correlation analysis showed there is a significant and positive correlation between the stigma length and SER (r = 0.46). However, K68S (SER = 0.82%) is an exception, as it showed a similar stigma length to K64S (SER = 22.81%) (Figure 4A, Table 2). These findings suggest that the stigma length is a key factor affecting the SER but not the only determining factor in wheat TPSGMS lines.

K66S had a minimum stigma branch angle (SBA) of 60.09°, while K68S had a maximum SBA of 137.33°. The SBA of K64S was 111.12°, not significantly different from K41S (94.98°). It seems that the stigma branch angle does not affect the stigma exsertion rate because both the lowest and highest values of SBA come from the lines with a lower SER, namely K66S and K68S (Figure 4B).

The stigma stretch widths (SSWs) of K66S, K41S, K68S, and K64S were 3.71 mm, 4.76 mm, 5.89 mm, and 5.99 mm, respectively. K64S had the largest stigma stretch width, significantly larger than K41S and K66S. Sterile lines with stigma exsertion seem to have a bigger stigma stretch width (Figure 4C).

The stigma hairbrush length (HBL) of K66S (0.68 mm) was significantly shorter than that of K41S (0.92 mm), K64S (0.91 mm), and K68S (1.02 mm). K41S, K64S, and K68S showed no significant differences in HBL, indicating that the stigma hairbrush length has no influence on stigma exsertion (Figure 4D).

The correlation analysis showed that the stigma stretch width had a positive but not significant correlation with the SER. In addition, the three sterile lines with stigma exsertion, namely K41S, K64S and K68S, had significantly longer stigma hairbrush lengths than K66S, but the correlation between the hairbrush length and SER did not reach a significant level (Table 2).

### 2.3. The Glume Characteristics of the Sterile Line

Each sterile line displayed a different glume/lemma shape during the flowering stage. The differences in the glume width, glume length among the four sterile lines and the glume morphology are displayed in Figure 5.

The differences in glume width (GLW) of four sterile lines were illustrated in Figure 6A.

K41S and K64S, with a higher SER of 21–22%, showed narrower glumes, which were 3.47 mm and 3.58 mm, respectively, while K68S had a maximum glume width of 3.98 mm, significantly different from other lines (Figure 6A). The correlation analysis showed that the glume width has a significantly negative correlation with the SER (r = −0.46, *p* < 0.05, Table 3). The results demonstrate that a narrower glume is beneficial to a higher SER. Nevertheless, K66S also had a narrower glume, similar to K41S and K64S, but it did not show any stigma exsertion, which implies that the glume width is only one of the factors affecting stigma exsertion in sterile wheat lines.

As for the difference in glume length (GLL) among the four lines, K66S had a minimum glume length of 10.70 mm, significantly shorter than the other lines. K68S had the longest glume at 11.98 mm, but this was not significantly different to K41S and K64S (Figure 6B). The results indicate that the stigma exsertion rate is not affected by the difference in the glume length of sterile lines, which was also confirmed by the results from the correlation analysis; there was no significant correlation between the GLL and SER (Table 3).

When the glume length-to-width ratio (GLL/GLW) is considered (Figure 6C), it exhibits a similar trend to the variation in SER in the four sterile lines, i.e., K64S (3.33) > K41S (3.27) > K68S (3.02) > K66S (2.96). Meanwhile, it significantly and positively correlated with the SER (r = 0.67, *p* < 0.01, Table 3). The results suggest that wheat TPSGMS lines with a high SER seem to have slender glumes and florets.

### 2.4. Differences of Floral Structure between Florets with and without Stigma Exsertion in K41S and K64S

The SERs of K41S and K64S were about 21–22% (Table 1), indicating that most florets did not have stigma exsertion. As doubled haploid lines, all the florets of K41S or K64S had the same genotype, which means that the difference in stigma exsertion among the florets of K41S or K64S resulted from non-genetic factors. According to the results from the comparative observation, it was the significant changes in the stigma branch angle and stigma stretch width between florets with and without stigma exsertion that resulted in stigma exsertion or non-exsertion in florets from the same spike, which was mutually confirmed in K41S and K64S (Table 4). In K41S and K64S, the stigma branch angles of florets with stigma exsertion were 107.26°and 123.26°, respectively, which were increased by 26.02° and 26.92° compared with that of florets without stigma exsertion. Likewise, the stigma stretch widths of the florets with stigma exsertion were 5.34 mm and 6.56 mm, which were separately 1.23 mm and 1.04 mm wider than that of florets without stigma exsertion in K41S and K64S. The other tested floret traits showed no significant differences (Table 4). These results revealed that the decreased stigma branch angle significantly reduced the stigma stretch width, and consequently, the stigma was unable to stretch out from the glumes.

### 2.5. The Relationships among the OSSR with Other Attributes

The average total hybrid grains (THGs) per ten spikes were recorded for K41S and K64S, with 38.10 and 40.60 grains, respectively. On the contrary, 27.30 and 32.90 grains were achieved by K66S and K68S, respectively. Furthermore, the out-crossing seed setting rate (OSSR) of the florets with stigma exsertion was significantly higher than that of the florets without exsertion. The hybrid seed proportions on female parents, e.g., K41S and K64S, recorded 41.37% and 42.11% of the total tested spikes, respectively. However, K66S and K68S had values of 31.38% and 33.50%, respectively. The grains formed on lines such as K41S and K64S were 17.8 and 19.7 higher than the grains formed on florets with the stigma exsertion (GSE) trait, respectively, and the same trend was recorded for GSE/THG (Table 5).

Furthermore, the out-crossing ability (out-crossing seed setting rate with exserted stigma/out-crossing seed setting rate with non-exserted stigma) of the florets with an exserted stigma was about 3.17 times that of the florets with a non-exserted stigma in K41S and K64S (Table 5). Moreover, there were significant correlation between the out-crossing seed setting rate (OSSR) and the number of florets with exserted stigma, SER, the grains of florets with exserted stigma and the grains of stigma exserted florets accounting for the total grains, with correlation coefficient values of 0.94, 0.93, 0.94 and 0.93, respectively (Figure 7). These results revealed that the SER is an essential factor affecting the out-crossing seed setting rate in the wheat TPSGMS line.

## 3. Discussion

Understanding the floral architecture and morphological characteristics of stigma exsertion in wheat is of great significance in developing sterile lines with a high out-crossing ability and further investigating the genetic basis of stigma exsertion.

Both wheat and rice are self-pollinating crops. The stigma exsertion rate in rice was demonstrated to be a decisive factor in hybrid seed yield at least 30 years ago [39]. The total out-crossed seeds produced from the florets with an exserted stigma was as high as about 75% (57.6–88.8%). For every 1% increase in the SER, the out-crossing seed setting rate increased by 0.74–0.96%. Most practical sterile rice lines thus had an SER of 60–80% [40]. Furthermore, in hybrid rice production, the florets with an exserted stigma had a significantly higher stability in the out-crossing seed setting rate than in florets without stigma exsertion under less favorable pollination conditions [39,40]. In wheat, similar findings also confirmed that the out-crossing ability of florets with stigma exsertion is higher than that of those without stigma exsertion in wheat TPSGMS lines [28,29]. In this study, the out-crossing seed setting rate (OSSR) of K41S (41.81%) and K64S (42.53%) was significantly higher than that of K66S (31.43%) and K68S (33.50%) because the SER of K41S and K64S was more than 20 times as high as that of K66S and K68S. Furthermore, the out-crossing seed setting rate of stigma-exserted florets in K41S and K64S was 3.17 times as high as that of florets without stigma exsertion; stigma-exserted florets produced 47% and 49% of the total cross-pollinated seeds in K41S and K64S, respectively, though these florets only accounted for 22–23% of the total florets. Yang et al. also reported similar results in photo-thermo-sensitive genic male sterile wheat [29]. Therefore, developing and utilizing sterile lines with a higher SER could be a promising approach to further raising the hybrid seed yield in wheat.

Stigma and glume properties synergistically determine the SER in rice [41,42,43,44,45,46]. Rice germplasm with high stigma exsertion is generally characterized by a slender spikelet (high ratio of spikelet length/spikelet width) and slender stigma (long stigma) [32,41]. In this study, K64S had the longest stigma among four TPSGMS lines and the highest SER (22.81%), and the stigma length was positively correlated with the SER. Meanwhile, even though K41S was shorter in stigma length than K64S, it also possessed the same level of SER (21.87%) as K64S because it had the narrowest glume. In addition, the glume length-to-width ratio was significantly and positively correlated with the SER (r = 0.60), suggesting that a slender floret in wheat TPSGMS lines could be beneficial to stigma exsertion; similar results were also reported in rice [42,47,48]. Generally, it seems that a long stigma combined with a slender glume is the basic characteristic of stigma exsertion in sterile wheat lines.

The SER is a complex trait that can be affected by genetics, environmental conditions, and plant growth regulators in rice [49,50,51,52,53]. A similar scenario also occurred in the wheat TPSGMS line, e.g., in hybrid seed production; the SER of K64S was about 15% under high-temperature and low-humidity conditions, but it could reach more than 25% under low-temperature and high-humidity conditions (unpublished data). Furthermore, when K64S was put in a climate chamber after heating (20 °C, photoperiod at 14 h and 12,000 Lux, relative humidity at 65%), the SER could even reach up to 95% (unpublished data). Therefore, further studies are necessary to understand how temperature and humidity affect the stigma branch angle and stigma stretch width, and consequently change the SER in sterile wheat lines like K41S and K64S, which is crucial for developing a sterile line with a high SER and the efficient production of hybrid seed in wheat.

## 4. Materials and Method

### 4.1. Wheat Materials

Four thermo-photo sensitive genic male sterile (TPSGMS) lines, namely K41S, K64S, K66S and K68S, as well as one restorer line, Yunmai112, were used in this research. All these lines were doubled haploids and developed by the Food Crops Research Institute, Yunnan Academy of Agricultural Sciences, China (Table 6). According to previous observations, K41S and K64S possess obvious stigma exsertion, and K66S and K68S have few or no exserted stigma.

### 4.2. Planting of Wheat Materials

Experiments were performed at the Experimental Farm of Songming county (25°35′ N, 102°98′ E, altitude 1997 m), Kunming, Yunnan province, China. Based on a recent 5-year observation, the optimum sowing date was 16 October for the sterile lines to show complete sterility. In this study, three sowing dates at seven-day intervals were used for the four sterile lines (female) and the restorer/pollinator Yunmai112 in two growing seasons of 2021/2022 and 2022/2023; this included 9 October, 16 October and 23 October 2021 and 2022, respectively. More details of the natural conditions of the Experimental Farm were described by Li et al. [36]. Each sterile wheat line consisted of five rows planted at 1 m length and 0.25 m spacing, and they were surrounded by six rows of the pollinator for the test of their out-crossing ability.

### 4.3. Sterility Evaluation of Sterile Lines

The sterility assessment of four sterile lines under three sowing times was conducted during the flowering stage. When the wheat anther grew light yellow or 50% of the glumes were opening, ten anthers from each sterile line were randomly collected for fertility examination with 1% iodine–iodide kalium (I_2_-KI) solution, then inspected under the microscope to identify and record the pollen fertility.

### 4.4. Measurement of Stigma and Glume Traits

During the flowering stage, at least ten spikes of each sterile line were randomly selected to measure the stigma and glume traits, such as the stigma exsertion, glume length and glume width. The stigma exsertion rate (SER) is calculated as follows:$$SER=\frac{Number\;of\;florets\;with\;exserted\;stigma}{Total\;number\;of\;observed\;florets}\times 100$$

In addition, five basal florets in the middle part of each spike were used to separate the pistils from glumes carefully. Photos of the stigma and ovary were then taken under a stereomicroscope (Leica M205C, Germany) to investigate or measure the floral traits using ImageJ 1.8.0 software (https://imagej.nih.gov/ij/, accessed on 1 December 2023). The stigma length (SL), stigma stretch width (SSW), stigma branch angle (SBA), and stigma hairbrush length (HBL) were separately measured as illustrated in Figure 8A. The glume length (GLL) and glume width (GLW) were measured as illustrated in Figure 8B.

### 4.5. Survey of Out-Crossing Ability in Four Sterile Line

The same ten spikes of each sterile line for SER measurements were used to estimate the out-crossing ability by investigating the out-crossing seed setting rate (OSSR) after the harvest. And the grains from florets with an exserted stigma (GFES) were recorded to evaluate the out-crossing ability of stigma-exserted florets. In addition, another ten spikes from each sterile line were bagged before flowering to calculate the number of self-fertilized seeds after harvest. The OSSR and GFES/TG are calculated as follows:$$OSSR(\%)=\frac{Total\;grains\;number}{Total\;florets\;number}\times 100$$
$$GFES/TG(\%)=\frac{Total\;number\;of\;grains\;from\;exserted\;stigma}{Total\;grains\;number}\times 100$$

### 4.6. Statistical Analyses

IBM SPSS Statistics [54] was used for ordinary data analysis.

## 5. Conclusions

The out-crossing ability or out-crossing seed setting rate (OSSR) of wheat TPSGMS lines is directly related to the yield and cost of hybrid seed production. In the present study, the stigma exsertion in wheat is synergistically determined by specific stigma and glume characteristics, especially the stigma length, glume width, and glume length–width ratio, which significantly correlated with the stigma exsertion rate. Moreover, for a sterile line with stigma exsertion, the exsertion rate was mainly affected by differences in the stigma branch angle and stigma stretch width among florets, which results from the interaction between the genotype and environment. Therefore, further research is required to investigate how genotype and environmental factors affect the stigma exsertion rate in sterile line and hybrid seed production. Furthermore, we also demonstrate that the out-crossing ability of florets with stigma exsertion in wheat TPSGMS lines is much better than that of florets without stigma exsertion. These findings increase our understanding of the morphological characteristics of stigma exsertion in sterile wheat lines, which will be beneficial for further genetic studies and the improvement of sterile wheat lines with stigma exsertion, as well as for cost-effective hybrid seed production.

## Figures and Tables

**Figure 1 plants-13-02267-f001:**
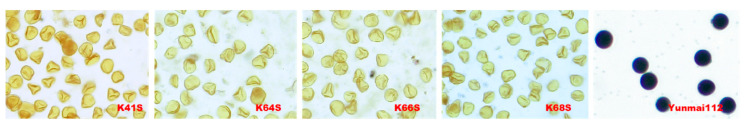
Microscope inspection of pollen samples’ shape and color after staining with 1% I_2_-KI solution, indicating the sterility of four sterile lines and fertility of Yunmai 112.

**Figure 2 plants-13-02267-f002:**
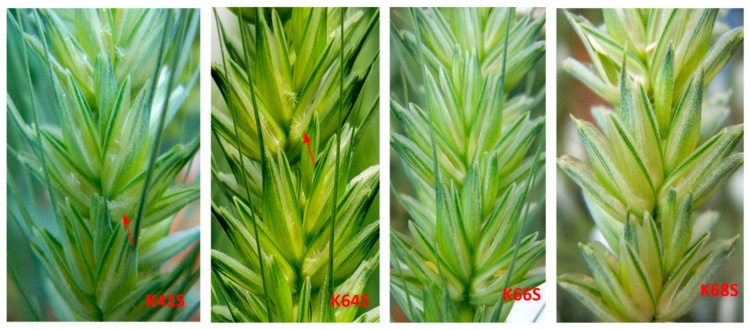
Stigma exsertion in four TPSGMS lines tested on three sowing dates in both seasons, and the red arrows show the stigmas stretching out from the glumes.

**Figure 3 plants-13-02267-f003:**
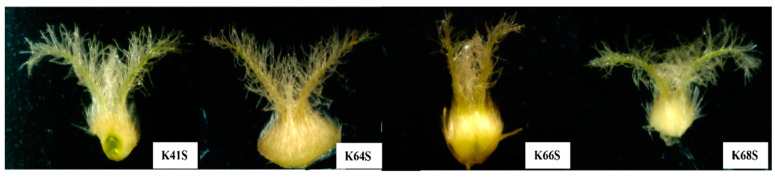
Morphological characteristics of the ovary and stigma in four sterile lines.

**Figure 4 plants-13-02267-f004:**
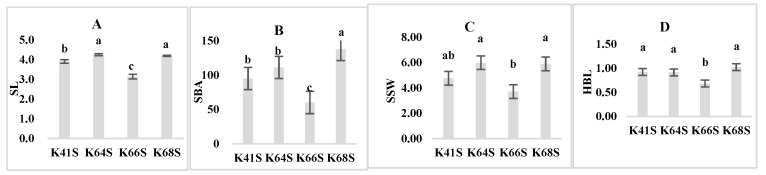
Stigma parameters of the four sterile lines: SL, stigma length (**A**); SBA, stigma branch angle (**B**); SSW, stigma stretch width (**C**); and HBL, stigma hairbrush length (**D**). Bars or columns with the same letter are not significantly different based on the least significant difference (LSD) at *p* ≤ 0.05.

**Figure 5 plants-13-02267-f005:**
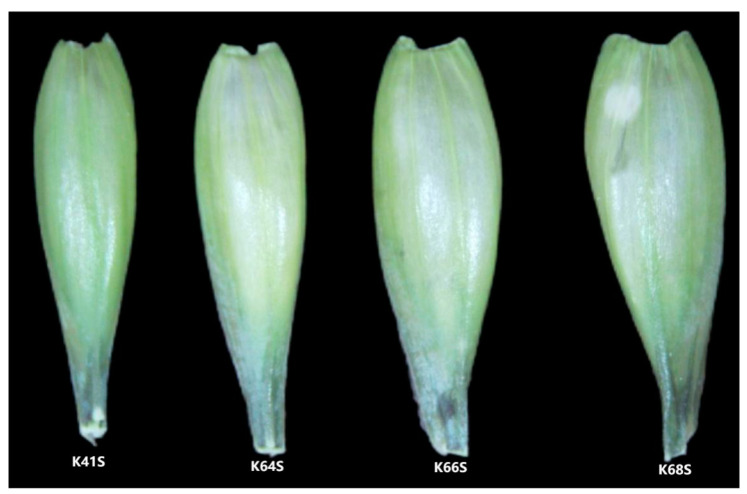
Morphological characteristics of glumes from four sterile lines during the flowering stage.

**Figure 6 plants-13-02267-f006:**
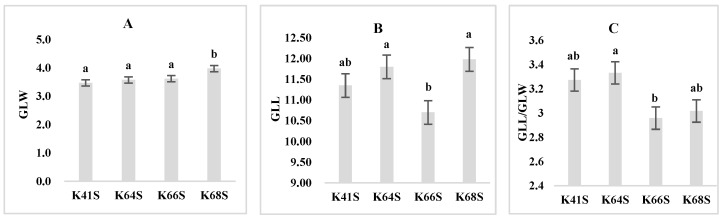
Glume characteristics of the four sterile lines: GLW, Glume width (**A**); GLL, Glume length (**B**); and GLL/GLW, the glume length-to-width ratio (**C**). Bars or columns with the same letter are not significantly different based on the least significant difference (LSD) at *p* ≤ 0.05.

**Figure 7 plants-13-02267-f007:**
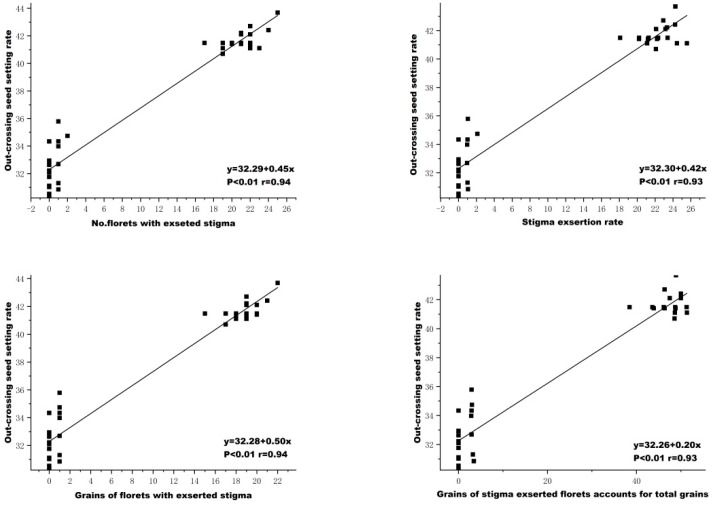
Correlation analysis between out-crossing seed setting rate and the traits of an exserted stigma.

**Figure 8 plants-13-02267-f008:**
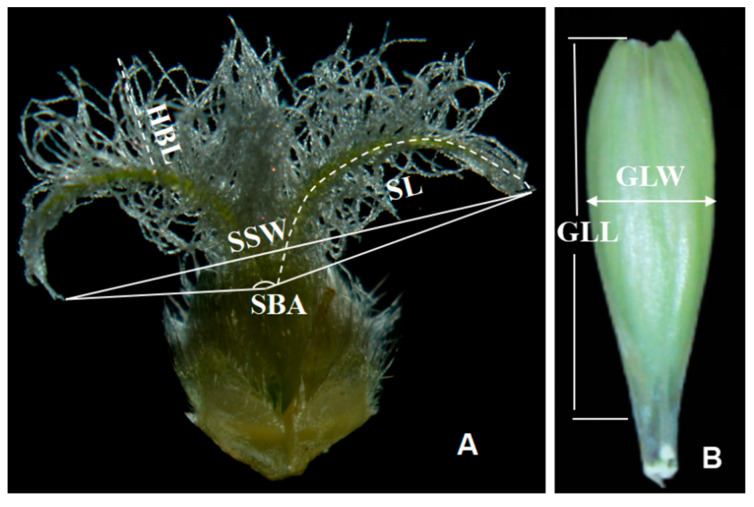
Sketch map of the stigma length (SL), stigma stretch width (SSW), stigma hairbrush length (HBL), stigma branch angle (SBA), glume length (GLL), and glume width (GLW) of sterile lines for measurement. (**A**) stigma and ovary; (**B**) glume.

**Table 1 plants-13-02267-t001:** Mean performance of the four sterile lines regarding the stigma exsertion rate.

Sterile Lines	TFN	FESN	SER
K41S	92.10 ± 3.60 b†	20.10 ± 1.73 b	21.87 ± 2.20 a
K64S	96.40 ± 3.66 a	22.00 ± 1.49 a	22.81 ± 1.09 a
K66S	87.00 ± 3.83 c	0.00 c	0.00 b
K68S	98.20 ± 3.46 a	0.80 ± 0.63 c	0.82 ± 0.66 b

† Mean values within the same trait column with the same letter are not significantly different based on the least significant difference (LSD) at *p* ≤ 0.05. TFN, total floret number; FESN, florets with exserted stigma numberer; SER, stigma exsertion rate.

**Table 2 plants-13-02267-t002:** The correlation among the SER and stigma characteristics of the four sterile lines.

Triats	SL	SBA	SSW	SHBL	SER
Stigma length (SL)	1.00	0.76 **	0.74 **	0.68 **	0.46 *
Stigma branch angle (SBA)	0.76 **	1.00	0.83 **	0.83 **	0.09
Stigma stretch width (SSW)	0.74 **	0.83 **	1.00	0.57 **	0.27
Stigma hairbrush length (HBL)	0.68 **	0.84 **	0.57 **	1.00	0.21
Stigma exsertion rate (SER)	0.46 *	0.09	0.27	0.21	1.00

* and ** indicate significant differences at the 0.05 and 0.01 levels.

**Table 3 plants-13-02267-t003:** The correlation among the SER and glume phenotype of the four sterile lines.

Triats	GLW	GLL	GLL/GLW	SER
Glume width (GLW)	1.00	0.40	−0.67 **	−0.46 *
Glume length (GLL)	0.40	1.00	0.41	0.20
Length-width ratio of glume (GLL/GLW)	−0.67 **	0.41	1.00	0.60 **
Stigma exsertion rate (SER)	−0.46 *	0.20	0.60 **	1.00

* and ** indicate significant differences at the 0.05 and 0.01 levels.

**Table 4 plants-13-02267-t004:** Comparison of floral traits between florets with and without stigma exsertion in K41S and K64S.

Lines	Stigma Status	SBA	SSW	HBL	SL	GLW	GLL
K64S	Florets with exserted stigma	123.26 ± 7.94 b	6.56 ± 1.02 b	0.92 ± 0.16 a	4.26 ± 0.69 a	3.60 ± 0.19 a	11.82 ± 0.16 a
	Florets without exserted stigma	96.34 ± 6.79 a	5.52 ± 0.86 a	0.91 ± 0.13 a	4.25 ± 0.52 a	3.58 ± 0.11 a	11.79 ± 0.13 a
K41S	Florets with exserted stigma	107.26 ± 6.83 b	5.34 ± 1.11 b	0.91 ± 0.10 a	3.92 ± 0.41 a	3.49 ± 0.22 a	11.37 ± 0.19 a
	Florets without exserted stigma	81.24 ± 5.66 a	4.13 ± 0.89 a	0.91 ± 0.09 a	3.91 ± 0.33 a	3.49 ± 0.20 a	11.36 ± 0.21 a

SL, stigma length; SBA, stigma branch angle; SSW, stigma stretch width; HBL, stigma hairbrush length; GLW, Glume width; GLL, Glume length; GLL/GLW, the glume length to width ratio. The same letter is not significantly different based on the least significant difference (LSD) at *p* ≤ 0.05. The data were collected randomly from the other ten spikes of K41S and K64S, and one basal floret with or without an exserted stigma at the middle party of each spike was used for the above trait testing.

**Table 5 plants-13-02267-t005:** Mean performance of the four sterile lines for the hybrid seed setting rate and indices.

Lines	K41S	K64S	K66S	K68S
TFN	92.10 ± 3.60 b	96.40 ± 3.66 c	87.00 ± 3.83 a	98.20 ± 3.46 c
THG	38.10 ± 1.66 c	40.60 ± 1.96 d	27.30 ± 1.16 a	32.90 ± 1.91 b
OSSR	41.37 ± 0.49 c	42.11 ± 0.72 c	31.38 ± 0.91 a	33.50 ± 1.58 b
GSE	17.80 ± 1.73 b	19.70 ± 1.49 c	0.00 a	0.70 ± 0.63 a
OSSR-ES	88.73 ± 1.83 b	89.60 ± 1.66 b	0.00 a	92.86 ± 4.74 b
OSSR-OES	28.14 ± 1.40 a	28.08 ± 1.00 a	31.39 ± 1.90 a	33.06 ± 1.69 a
GSE/TG	46.80 ± 3.85 c	48.53 ± 1.82 c	0.00 a	2.14 ± 1.49 b

TFN, total floret number; THG, total hybrid grains; OSSR, out-crossing seed setting rate; GSE, grains of florets with exserted stigma; OSSR-ES, out-crossing seed setting rate of florets with exserted stigma; OSSR-OES, out-crossing seed setting rate of florets with non-exserted stigma; GSE/TG, grains of the florets with exserted stigma accounts for total grains. Mean values within the same trait line with the same letter are not significantly different based on the least significant difference (LSD) at *p* ≤ 0.05.

**Table 6 plants-13-02267-t006:** Brief description of the wheat materials used in the study.

Lines	Pedigree	Stigma Exsertion	Percent of Glume Openness	Plant Height (cm)
K41S	K78S/14Y6-438	Exsert	90%	65
K64S	K78S/14Y6-172	Exsert	90%	70
K66S	K78S/14Y6-172	Non-exsert	85%	60
K68S	K456S/14Y6-438	Trivial exsert	85%	75
Yunmai112	An elite plant from population improvement	-	-	100

## Data Availability

All data generated or analyzed during this study are included in this published article and are available from the corresponding authors upon reasonable request.

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
