# Peer review of "Stigma and Glume Characteristics Synergistically Determine the Stigma Exsertion Rate in Thermo-Photo-Sensitive Genic Male Sterile Wheat"

_plants, 2024, doi:10.3390/plants13162267_

Round 1

Reviewer 1 Report

Comments and Suggestions for Authors

All measured data in the article were not given standard error. It should be given.

Author Response

Dear Reviewer,

We are thankful for the general comments on improving the quality of our manuscript. We have

worked to improve the manuscript as per your comments/suggestions. Please find a point-by-point response to your recommendations.

we appreciate your notes, and we have given the standard error in the revised version .

Reviewer 2 Report

Comments and Suggestions for Authors

This manuscript studied the relationship between the hybrid seed setting rate and the floral architecture relative to stigma exsertion.  Two materials were proved to be useful for the hybrid wheat. The results in this manuscript are useful for the hybrid wheat breeding. Some minor revisions are needed.

 1.     In the Abstract, what does mean “longer/exserted”? It means longer and exserted? Or it means longer or exserted?

2.     The style of the title of Table 3 should be revised. The “SAB”, “SSW”, “HBL” ect. In the Table 3 should be explained as note.

3.     In the reported studies, whether the seed setting rate of hybrid wheat is involved. If there are previous studies in this content, the results of this study can be compared with those of previous studies.

Author Response

Dear Reviewer,

Thank you for your valuable comments on our manuscript. We have worked to improve the manuscript by following suggestions and enhancing its quality. Please find below point-by-point responses to your various concerns.

 Comments and Suggestions for Authors

This manuscript studied the relationship between the hybrid seed setting rate and the floral architecture relative to stigma exsertion.  Two materials were proved to be useful for the hybrid wheat. The results in this manuscript are useful for the hybrid wheat breeding. Some minor revisions are needed.

  1. In the Abstract, what does mean “longer/exserted”? It means longer and exserted? Or it means longer or exserted?

Response point 1: Thank you very much for this comment. We improve that sentence to be as “These findings suggest that the HSSR increased by stigma exsertion rate combined with “

  1. The style of the title of Table 3 should be revised. The “SAB”, “SSW”, “HBL” ect. In the Table 3 should be explained as note.

Response point 2. We appreciate your note. We improve the title name and the table’s footnote by adding the traits abbreviations accordingly.

  1. In the reported studies, whether the seed setting rate of hybrid wheat is involved. If there are previous studies in this content, the results of this study can be compared with those of previous studies.

Response point 3: We appreciate your valuable suggestions. We added an explanation and comparison with the previous studies in the second paragraph (inserted in the discussion section)

Reviewer 3 Report

Comments and Suggestions for Authors

1.       The experimental design of this study is excellent and holds significant practical importance for wheat breeding.

2.       The preliminary preparation of experimental materials is commendable, and the selection of experimental germplasm was done with great care.

However, there are some issues that must be addressed before publication:

1.       Title:

The title should be revised to avoid phrases such as "such as."

2.       Figures:

2.1.      The numbering of figures should follow the order of appearance. Currently, Figure 1 is placed later in the document.

2.2.      The background of the images in Figure 2 is blue, indicating a lack of white balance during microscopic photography. The color temperature is incorrect, leading to color distortion. White balance calibration should be performed. If the authors are unsure how to address this, they should seek assistance from an experienced microscopy technician.

2.3.      Figure 3 is too blurry, making it difficult to discern details and text within the image. Additionally, the caption does not explain what the red arrows are pointing to.

2.4.      None of the graphs in Figure 5 include error bars. These should be added. Also, it is not indicated how many samples the values in Figure 5 are based on (N=?). The text in Figure 5A is too blurred to read.

2.5.      The color temperature in all images of Figure 6 is incorrect and should be corrected with proper white balance during photography. The text in Figure 6 is too small to read.

2.6.      Figure 7 has the same issues as Figure 5. Additionally, the font size in Figure 7C is not consistent with Figures 7A and 7B.

2.7.      Figures 9 and 10 are too blurry, and the text is completely unreadable.

2.8.      Overall, there are too many figures. The number of figures should be reduced by integrating and reformatting them. A reasonable number would be between 5-8 figures. For example, Figures 2 and 3 can be combined into one figure, Figures 4 and 6 can be merged into one figure based on layout, and Figures 5 and 7 can be combined into one figure.

2.9.      Once all the figures have been reordered and reformatted, the citations in the text should also be revised.

3.       Tables:

3.1.      The numbering of tables should follow the order of appearance in the text. Currently, Table 1 appears later in the document.

3.2.      Both Table 2 and Table 3 are missing standard deviation (SD) values. The values should be presented as mean±SD. It is also unclear how many biological replicates each data point represents (N=?).

3.3.      In lines 167-168, the note in Table 2 states, "Lowercase letters indicate significant differences at the 0.05 level, the same as below." The intended meaning is that different lowercase letters indicate significant differences (p < 0.05). This note should be rewritten for clarity. The same issue exists in table 3 as well.

4.       Language and Grammar:

4.1.      It is recommended that the manuscript undergo language editing to eliminate grammatical errors and improve the overall clarity and quality of expression.

Comments on the Quality of English Language

  It is recommended that the manuscript undergo language editing to eliminate grammatical errors and improve the overall clarity and quality of expression.

Author Response

Dear Reviewer,

Thank you for your valuable comments and suggestions for our manuscript. We have worked to improve the manuscript by following suggestions and enhancing its quality. Please find below point-by-point responses to your various concerns.

  1. The experimental design of this study is excellent and holds significant practical importance for wheat breeding.
  2. The preliminary preparation of experimental materials is commendable, and the selection of experimental germplasm was done with great care.

However, there are some issues that must be addressed before publication:

  1. Title:

Point 1: The title should be revised to avoid phrases such as "such as."

Response point 1: We appreciate your notes and improved the title as much as possible.

  1. Figures:

2.1.     Point 2.1: The numbering of figures should follow the order of appearance. Currently, Figure 1 is placed later in the document.

Response point 2.1: Thank you for your comment; we rearranged the figures and tables in the entire manuscript.

2.2.    The background of the images in Figure 2 is blue, indicating a lack of white balance during microscopic photography. The color temperature is incorrect, leading to color distortion. White balance calibration should be performed. If the authors are unsure how to address this, they should seek assistance from an experienced microscopy technician.

Response point 2.2. Thank you for this notation. We inserted a good-quality photo of this figure.

2.3.      Figure 3 is too blurry, making it difficult to discern details and text within the image. Additionally, the caption does not explain what the red arrows are pointing to.

Response point 2.3. Thank you for your valuable comments; we put the images into the Word files as JPG or picture, and when converting the file to PDF, it appeared as blurry. So, we will send the original images and figures as separate files to the journal.

And 

In the revised manuscript version of the PDF, ensure it is saved in good quality, especially the figures

And

We add “and the red arrows show the exited stigmas from the glumes.” In the caption to explain the role of red arrows.

2.4.      None of the graphs in Figure 5 include error bars. These should be added. Also, it is not indicated how many samples the values in Figure 5 are based on (N=?). The text in Figure 5A is too blurred to read.

Response point 2.4: Thank you for your notes. We have improved the figure 5

And

 We add the following explanation of sample size in the caption of Figure 4: ‘Mean for selected ten spikes per sterile lines in the optimum sowing date in both seasons’.

And

We adjusted the font and size to keep the consistency of all figure parts and inserted in the manuscript.

2.5.      The color temperature in all images of Figure 6 is incorrect and should be corrected with proper white balance during photography. The text in Figure 6 is too small to read.

Response point 2.5: We appreciate your suggestions; this figure was replaced with a good photo to represent the morphological differences among the glume of genotypes.

2.6.      Figure 7 has the same issues as Figure 5. Additionally, the font size in Figure 7C is not consistent with Figures 7A and 7B.

Response point 2.6: Thank you for your notes. Furthermore, we used table instead of figure 7

2.7.      Figures 9 and 10 are too blurry, and the text is completely unreadable.

Response point 2.7. Thank you for your valuable comments; we used another figure instead of 9 and 10

 In the current revision round, we ensured that the PDF file of the manuscript had the figures in good quality

2.8.      Overall, there are too many figures. The number of figures should be reduced by integrating and reformatting them. A reasonable number would be between 5-8 figures. For example, Figures 2 and 3 can be combined into one figure, Figures 4 and 6 can be merged into one figure based on layout, and Figures 5 and 7 can be combined into one figure.

Response to point 2.8: We appreciate your comments and divided the results in this arrangement (show the stigma traits separated from the glume traits) to avoid the complexity in representing the results so the reader can easily find the needed information from the figures and text. We have deleted some figures, and there are 8 figures in the revised version.

2.9.      Once all the figures have been reordered and reformatted, the citations in the text should also be revised.

Response to point 2.9: We appreciate your comments and divided the results in this shape (show the stigma traits separated from the glume traits) to avoid the complexity in representing the results, so the reader can easily find the needed information from the figures and text.

The correct order of tables and figures in the results, materials, and methods was done.     

  1. Tables:

3.1.      The numbering of tables should follow the order of appearance in the text. Currently, Table 1 appears later in the document.

Response point 3.1: Thank you for your comment; we rearranged the figures and tables in the entire manuscript.

3.2.      Both Table 2 and Table 3 are missing standard deviation (SD) values. The values should be presented as mean±SD. It is also unclear how many biological replicates each data point represents (N=?).

Response point 3.2: we appreciate your notes, and we have given SD values.

And

We added the following phrase to the table caption ‘of selected ten spike samples from sowing dates and both seasons’ to explain the observations that were measured in Tables 1 & 2.

3.3.     In lines 167-168, the note in Table 2 states, "Lowercase letters indicate significant differences at the 0.05 level, the same as below." The intended meaning is that different lowercase letters indicate significant differences (p < 0.05). This note should be rewritten for clarity. The same issue exists in table 3 as well.

Response point 3.3: We adjusted the first table by removing the repetition and adding the definition to the footnote of Table 2.

  1. Language and Grammar:

4.1.      It is recommended that the manuscript undergo language editing to eliminate grammatical errors and improve the overall clarity and quality of expression.

Response 4.1:    We appreciate your recommendation; we improved the title, introduction results, discussion, and materials and methods, considering the English language, editing and rephrasing, and fixing the grammatical errors in the whole manuscript. This is to enhance the Quality of the English Language.